# The Classification of Farming Progress in Rice–Wheat Rotation Fields Based on UAV RGB Images and the Regional Mean Model

**Xiaoxin Song** [1,2], **Fei Wu** [1,2], **Xiaotong Lu** [1,2], **Tianle Yang** [1,2], **Chengxin Ju** [1,2], **Chengming Sun** [1,2,*] and **Tao Liu** [1,2]

1   Jiangsu Key Laboratory of Crop Genetics and Physiology/Jiangsu Key Laboratory of Crop Cultivation and Physiology, Agricultural College of Yangzhou University, Yangzhou 225009, China; MZ120201166@yzu.edu.cn (X.S.); yzuwufei2020@gmail.com (F.W.); luxiaotong516@gmail.com (X.L.); mz120180893@yzu.edu.cn (T.Y.); cxju@yzu.edu.cn (C.J.); tliu@yzu.edu.cn (T.L.)
2   Jiangsu Co-Innovation Center for Modern Production Technology of Grain Crops, Yangzhou University, Yangzhou 225009, China
*   Correspondence: cmsun@yzu.edu.cn

**Abstract:** Extraction of farming progress information in rice–wheat rotation regions is an important topic in smart field research. In this study, a new method for the classification of farming progress types using unmanned aerial vehicle (UAV) RGB images and the proposed regional mean (RM) model is presented. First, RGB information was extracted from the images to create and select the optimal color indices. After index classification, we compared the brightness reflection of the corresponding grayscale map, the classification interval, and the standard deviation of each farming progress type. These comparisons showed that the optimal classification color indices were the normalized red–blue difference index (NRBDI), the normalized green–blue difference index (NGBDI), and the modified red–blue difference index (MRBDI). Second, the RM model was built according to the whole-field farming progress classification requirements to achieve the final classification. We verified the model accuracy, and the Kappa coefficients obtained by combining the NRBDI, NGBDI, and MRBDI with the RM model were 0.86, 0.82, and 0.88, respectively. The proposed method was then applied to predict UAV RGB images of unharvested wheat, harvested wheat, and tilled and irrigated fields. The results were compared with those obtained with traditional machine learning methods, that is, the support vector machine, maximum likelihood classification, and random forest methods. The NRBDI, NGBDI, and MRBDI were combined with the RM model to monitor farming progress of ground truth ROIs, and the Kappa coefficients obtained were 0.9134, 0.8738, and 0.9179, respectively, while traditional machine learning methods all produced a Kappa coefficient less than 0.7. The results indicate a significantly higher accuracy of the proposed method than those of the traditional machine learning classification methods for the identification of farming progress type. The proposed work provides an important reference for the application of UAV to the field classification of progress types.

**Keywords:** UAV RGB image; regional mean model; color index; rice–wheat rotation field; farming progress; classification; precision agriculture

## 1. Introduction

Farmland plays an important role in the material basis of human survival and development. With the development of precision agriculture, the determination of farmland types and the types of crops grown has become a basic task [1]. The Large Area Crop Inventory Experiment (LACIE) of the United States and the Monitoring Agriculture with Remote Sensing (MARS) program of the European Union both include the task of remote sensing surveying of farmland [2]. In 2003, the Russian Ministry of Agriculture also established a national agricultural monitoring system [3]. This system acquires information on farmland area, maps of farmland use status, and crop growth status. These systems are based on the intra-annual variation process of moderate resolution imaging spectroradiometer

(MODIS) vegetation indices for the estimation and analysis of crop and farmland areas. In the African region, Egypt used multi-temporal MODIS data and time series analysis methods to analyze the satellite images of irrigated regions at different stages [4]. This program has achieved the goals of farmland area surveying and monitoring of dynamic changes throughout Egypt. The rice–wheat rotation field is one of the main farming methods in China, with 4.8 million hectares of land under cultivation [5]. However, Chinese farmland areas are relatively small and fragmented [6]. In the study of the relationship between farmland fragmentation and agricultural production cost, many scholars use the average plot area, the number of plots, and the average plot distance to quantitatively reflect farmland fragmentation [7–9]. According to statistics, in China, the average arable land area of each household is only 0.58 hm$^2$, the number of plots in each household is as high as 5.34, the average plot area is only 0.11 hm$^2$, and the average distance from home is 1.06 km [10]. Thus, monitoring farming progress in rice–wheat rotation fields is an essential part of ensuring grain production. At present, this monitoring is generally conducted by manual field surveys that are time-consuming and inefficient. At the same time, inadequate monitoring of the farming conditions during the wheat harvesting stage could directly affect wheat harvesting and rice planting, resulting in a decrease in farmland utilization.

Remote sensing technology has been widely used for crop growth monitoring due to its advantages of high efficiency and accuracy [11–13]. Based on the sensing distance, the most commonly used remote sensing platforms can be classified into satellite remote sensing platforms, unmanned aerial vehicle (UAV) remote sensing platforms, and ground platforms [14]. Compared with the latter two, satellite remote sensing platforms can cover a larger area and have been used to assess various crop growth parameters such as leaf area index [15], cover [16], biomass [17], chlorophyll, nitrogen [18,19], and yield [20]. However, satellite remote sensing platforms produce only low-resolution images, have a long access cycle, and are affected by the weather [21]. These drawbacks make them unsuitable for farming progress monitoring of small-scale rice–wheat rotation farmlands. In the ground platforms, various mobile vehicles or portable instruments are exploited to obtain information about farmland or crops. Although the ground platforms can provide images with high resolution, their relatively low working efficiency cannot meet the requirements of fast and efficient monitoring of farming progress.

In recent years, UAV remote sensing technology has developed rapidly, which is making up, gradually, for the shortcomings of satellite remote sensing platforms and ground remote sensing platforms. UAV remote sensing platforms include fixed-wing and multirotor configurations. Fixed-wing UAVs have the advantages of fast flight, high flight efficiency, long endurance time, large payload capacity, and a high flight altitude [22,23]. However, they have certain requirements for taking off and landing, they cannot hover, and they cause blurred images because of their high-speed shooting [24]. Multirotor UAVs have a simple structure, the ability to hover, and modest requirements for taking off and landing, but they possess a slow flight speed, short endurance time, a low flight altitude, and small payload volume [24]. UAV remote sensing platforms can acquire high-resolution crop canopy images. Moreover, they can carry different image sensors for assessing various crop growth parameters, including biomass [25], yield [26], LAI [27], chlorophyll [28], and nitrogen [29]. Makanza et al. [30] obtained visible light images by UAV and successfully achieved the estimation of canopy cover in maize fields. Roth [31] et al. realized the estimation of soybean LAI based on UAV RGB images, and the R$^2$ reached 0.92, laying a foundation for automatic LAI evaluation. Their results provide a theoretical basis for crop growth monitoring.

The above-mentioned studies demonstrated the potential of UAV remote sensing platforms for crop growth monitoring in farmland. It has been shown that crop classification using the UAV images analysis technique is an effective tool [32]. Wilke et al. [33] classified wheat with different lodging degrees by using a UAV-based canopy height model, the classification accuracy of R$^2$ reached 0.96. Wang et al. [34] extracted vegetation information by calculating various vegetation indices. The most accurate index was found to be VDVI

(the visible-band difference vegetation index) that could reach an accuracy of more than 90%; however, they didn't classify different field types. The above-mentioned studies mostly used single extraction methods such as color, texture features, or color indices; however, there are very few studies in the literature using multi-feature fusion and different extraction methods for classification and comparison.

Furthermore, UAV remote sensing images contain high-throughput information that could be used to characterize changes in farming progress, but most investigations only used the high-throughput information to monitor crop growth, and few studies were oriented toward farmland research to monitor farming progress during harvest. Roth et al. [35] successfully predicted yield and protein content through the dynamics of soybean vegetative growth based on hyperspectral images. They provided strong evidence for UAV remote sensing in crop physiological analysis. However, their research focused on the crop itself, with little mention of monitoring soybean fields to guide farm operations. Hassan et al. [36] realized the rapid monitoring of NDVI in the wheat growth cycle by using the multi-spectral UAV platform, which can realize the prediction of grain yield. Their research focuses on the flowering stage and filling stage of wheat growth stage, but the growth and field monitoring of wheat harvest stage are seldom mentioned. Field monitoring during the wheat harvest stage is important for guiding farmland operation of this stage. This indicates the importance of conducting researches on farming progress monitoring in rice–wheat rotation fields based on UAV remote sensing platforms during the wheat harvest stage.

Therefore, the objective of this study was to explore the optimal classification method for crop progress classification of rice–wheat rotation fields at the harvest stage. This classification method can effectively improve rice–wheat rotation farmland monitoring practices for farmers and provide a new idea for the visible image classification based on UAV. In this paper, we present a method to effectively extract information on farming progress using UAV RGB images and the RM model. The remainder of the paper is organized as follows. Section 2 describes the proposed methods and the corresponding materials being used. Section 3 describes the performance of the proposed method as well as the comparison with traditional machine learning methods. Section 4 concludes this paper and provides further discussion.

## 2. Materials and Methods

### 2.1. Image Acquisition and Preprocessing

We used DJI Inspire 2 UAV (Figure 1) as the platform to collect RGB images. The main technical parameters of the platform are as follows: hover accuracy: vertical: ±0.5 m, horizontal: ±1.5 m. The UAV can fly for 23 min with a pair of batteries and can hover and fly stably indoors and outdoors. The UAV was equipped with a Zenmuse X7 lens. We collected the UAV RGB images from the experimental field of Yangzhou University on 28 May 2020. The weather on the measurement day was sunny, and the UAV flew for a short time from 11:00 to 13:00 to ensure sufficient and stable sunlight. The UAV is battery powered with autonomy for about two hours. The shooting angle was perpendicular to the ground, and the flight height was about 20 m, as shown in Figure 2. The test field is located in Wenhui Road Campus of Yangzhou University ($32.39°$ N and $119.42°$ E, in 2020), with a total area of 3.76 hectares. The type of soil in this area is loamy soil. The main crops in the tested fields in May and June were wheat (cultivars: Yangmai-16 and Yangmai-23) and rice (cultivars: Suxiu-867 and Naneng-9108).

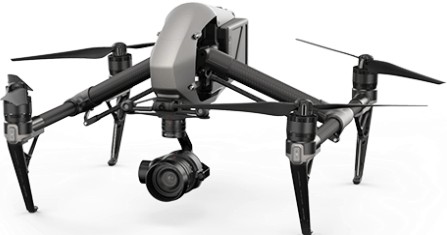

**Figure 1.** DJI Inspire 2 drone.

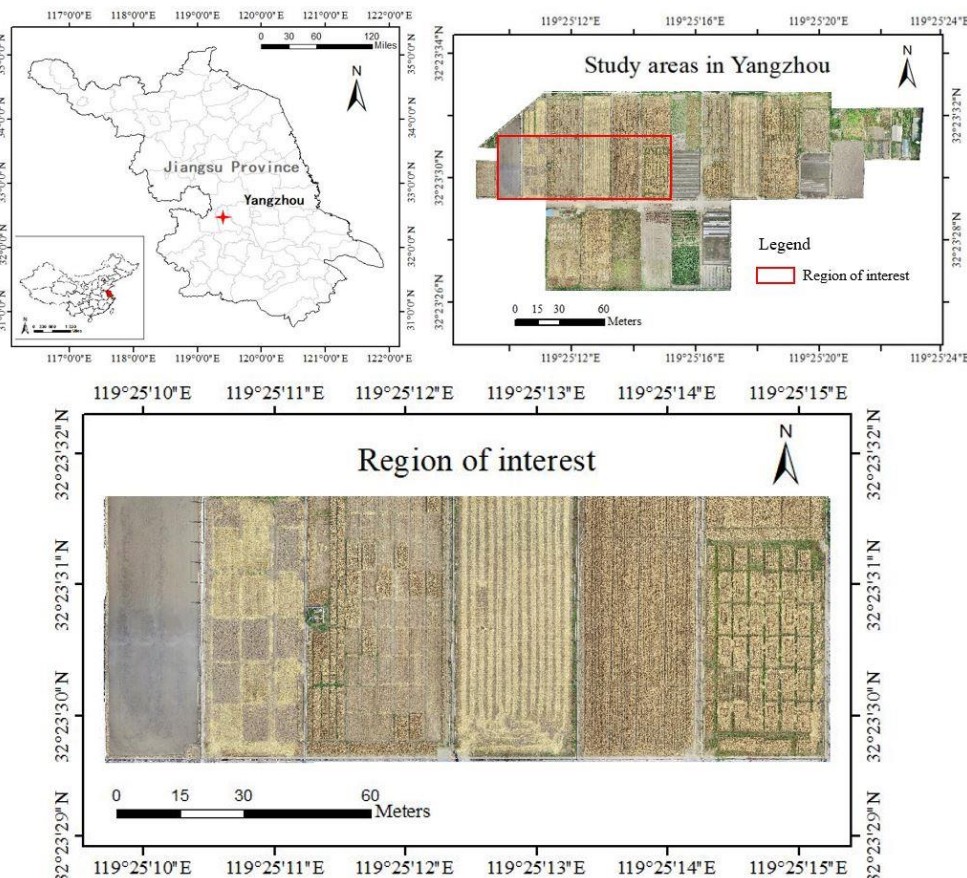

**Figure 2.** Visible light images of the study region.

We used the image stitching method and obtained RGB images to synthesize the whole test field orthophoto. Following the processing procedure of Lu et al. [37]; image alignment, georeferencing, mosaicking, dense point cloud construction, orthoimage generation, and calibration were conducted using PhotoScan Professional software (Agisoft LLC, St. Petersburg, Russia) and ENVI/IDL software (Harris Geospatial Solutions, Inc., Broomfield, CO, USA) based on the GPS location and camera internal parameters from each aerial image. Due to a large amount of data, to avoid the influence of blocking factors such as buildings on the classification of farming progress monitoring, only a part of the area located in the mid-western part of the flight area was selected as the region of interest in this study. As shown in Figure 2, the region of interest should include the type of farming progressing from the wheat harvest stage to the rice planting stage. The region of interest we selected here contained 16611 × 5962 pixels, and the corresponding grayscale values (red, green, blue) were stored in TIFF format, where each value consists of 8 bytes.

### 2.2. Color Index Construction and Selection

We calculated and analyzed the information from the RGB images and features of the color index to identify the key feature parameters that can be used to distinguish different farming progress types. Through visual interpretation and field investigation, samples were randomly selected from four agricultural progress types, as shown in Figure 3. As seen in the images, unharvested wheat fields (Figure 3a) are golden yellow, harvested wheat fields (Figure 3b) are light yellow due to wheat stubble, tilled wheat fields (Figure 3c) are grayish brown, and irrigated fields (Figure 3d) are gray. The color differences in these areas result from the differences in the phenotypic materials and the structures of the fields.

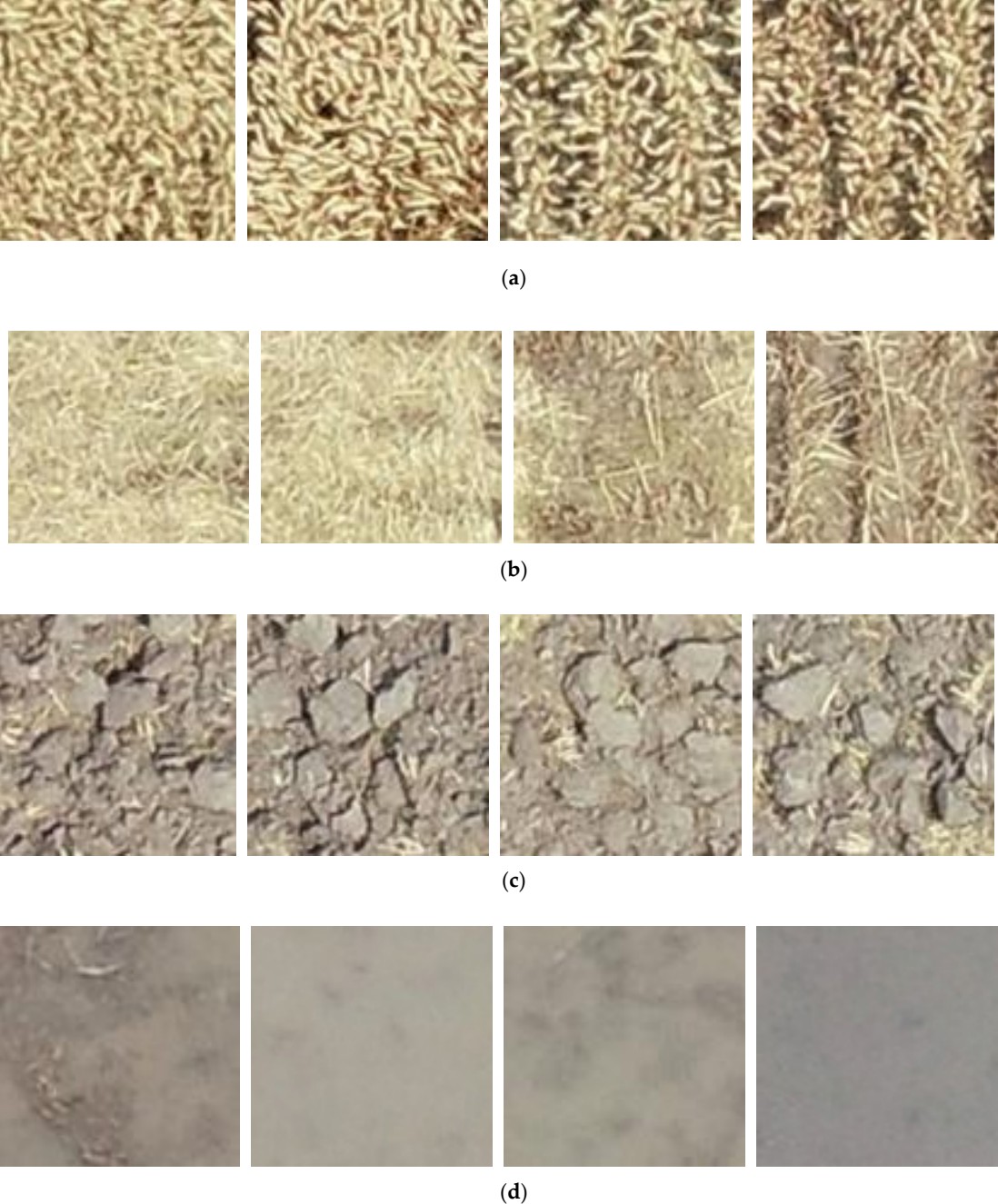

**Figure 3.** Some typical samples of farming progress in the rice-stubble wheat region. (**a**) Unharvested wheat fields. (**b**) Harvested wheat fields. (**c**) Tilled fields. (**d**) Irrigated fields.

Different types of farming progress in rice–wheat rotation fields clearly appear in the visible light band. After all essential image processing, digital number (DN) values of *R*, *G*, and *B* channels for each plot were extracted from each image using the region of interest (ROI) tool in ENVI/IDL. The DN values of RGB channels in remotely sensed RGB images can quantitatively reflect the radiance and reflectance characteristics in the visible spectrum of the crop canopy [38]. Normalized DNs (*r*, *g*, and *b*, Equations (1)–(3)) were reported to have superior capacity for vegetation estimation compared to original RGB DN values [39]. In this study, we will construct a further method based on the data in three visible light bands extracted by UAV; we will further analyze the regular changing in each band to create and select indices. The extraction of color features is realized by ENVI5.3.

$$r = R/(R + G + B), \tag{1}$$

$$g = G/(R + G + B), \tag{2}$$

$$b = B/(R + G + B), \tag{3}$$

where *R*, *G*, and *B* are the DNs of the red, green, and blue bands of the ROIs of each plot in the UAV-based images, respectively.

The color feature extraction is illustrated in Figure 4. The RGB values of different farming progress types were different. The RGB values of four different types of farming progress all decreased step by step with the R, G, and B channels, while the degree of changing for different farming progress types varied. The unharvested wheat fields showed the greatest change in the RGB channels, and the irrigated fields showed the least change in the RGB channels. Therefore, we constructed color indices based on this changing pattern to classify the four types of farming progress fields. The constructed color indices were NRBDI (normalized red-blue–difference index) and MRBDI (modified red–blue difference index). We also tried to use the color indices constructed by previous authors such as NGBDI and NGRDI as shown in Table 1.

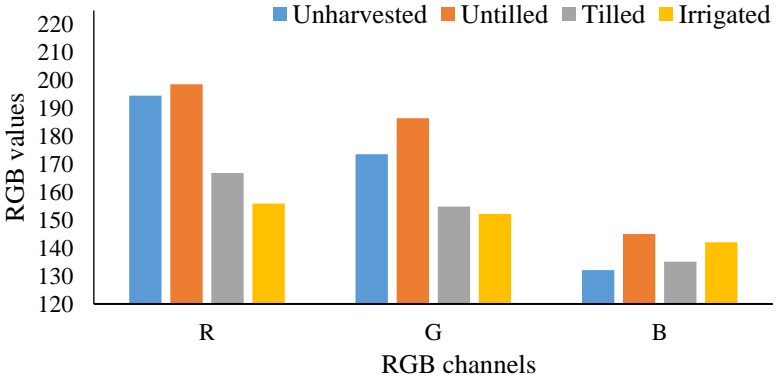

**Figure 4.** RGB values for four types of farming progress.

**Table 1.** Color indices formula.

| Characteristics Index | Index Name | Formula | References |
|---|---|---|---|
| NRBDI | Normalized Red–Blue Difference Index | $(r - b)/(r + b)$ | |
| NGBDI | Normalized Green–Blue Difference Index | $(g - b)/(g + b)$ | [40] |
| NGRDI | Normalized Green–Red Difference Index | $(g - r)/(g + r)$ | [40] |
| MRBDI | Modified Red–Blue Difference Index | $(r^2 - b^2)/(r^2 + b^2)$ | |

By recombining red, green, and blue images, a grayscale image resulted with perceptual difference in brightness for different land feature types [40]. We screened the indices, which were selected or constructed by visual interpretation to achieve accurate classification. The selected samples of different farming progress types were also analyzed to obtain

the characteristic values of different farming progress types to analyze the feasibility of the index classification.

### 2.3. Model Construction

By means of visual observation combined with field investigation, samples were randomly selected from each agricultural progress type to cover most features of the same agricultural progress type. Ten samples of unharvested wheat, harvested wheat, tilled, and irrigated fields were selected by visual inspection. We analyzed these samples and the corresponding grayscale images for these samples. Then, we constructed the model based on the color indices after analyzing the histograms of the indices.

The histograms of the grayscale values of the three color indices on the farmland with different types of farming progress are shown in Figure 5. NRBDI is mainly distributed around 0.2 on unharvested wheat fields, around 0.15 on harvested wheat fields, around 0.1 on tilled fields, and below 0.07 on flooded fields. The NGBDI is mainly distributed around 0.17, 0.13, 0.075, and below 0.053, respectively. The MRBDI is distributed around 0.4, 0.3, 0.2, and below 0.14, respectively. The distributions are all convex arcs, with lower proportions of values at both ends and more concentrated in the middle.

The distribution of the constructed indices was calculated from the above analysis. Although there is less distribution at both ends, there is still a certain amount of overlap. To be able to classify fields in their entirety to meet the needs of agricultural operations, we considered reducing or even eliminating the influence of the values at both ends of the distribution on the classification. Therefore, we created a new method called regional mean model. The rice–wheat rotation fields are rectangular, and according to the size of the fields, we choose a sample size of $100 \times 100$ pixels. The color index value at the $(i, j)$ point in the image is denoted as $A = f(x\_((i, j)))$, and the index range is set to 0–1. The average value of the color index is calculated for each sample, as given in Equation (4).

$$m = \frac{\sum_{i=1}^{i=100} \sum_{j=1}^{j=100} f\left(x_{(i,j)}\right)}{100 \times 100} \tag{4}$$

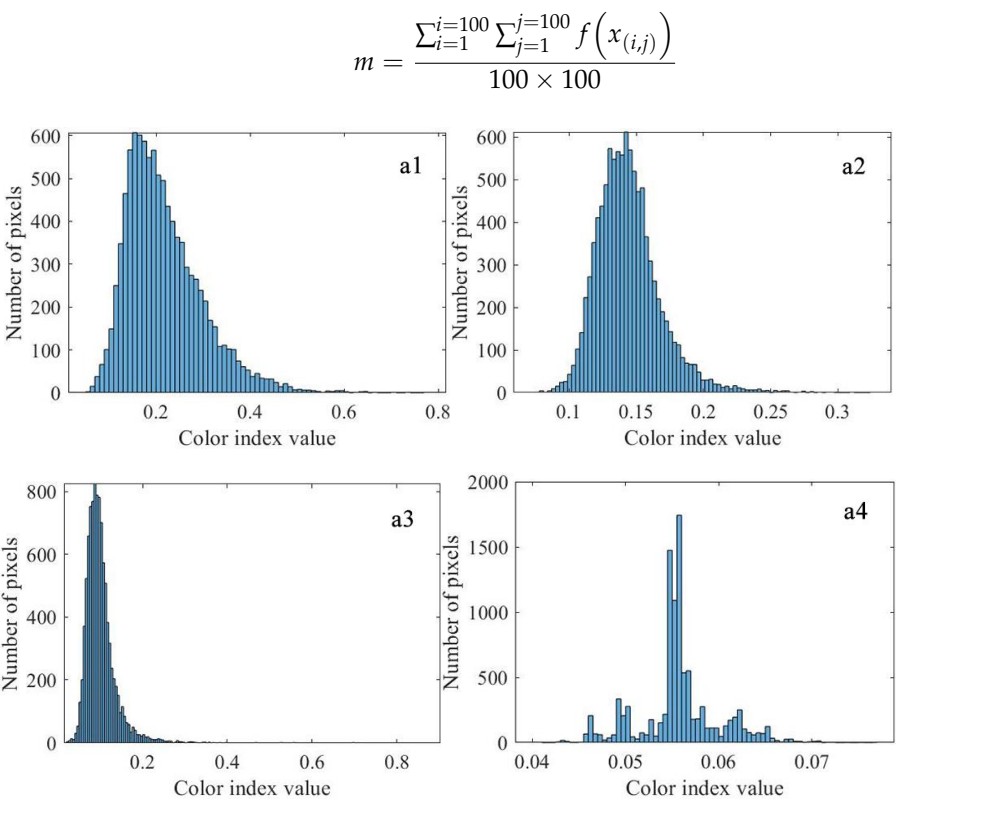

(**a**)

**Figure 5.** *Cont.*

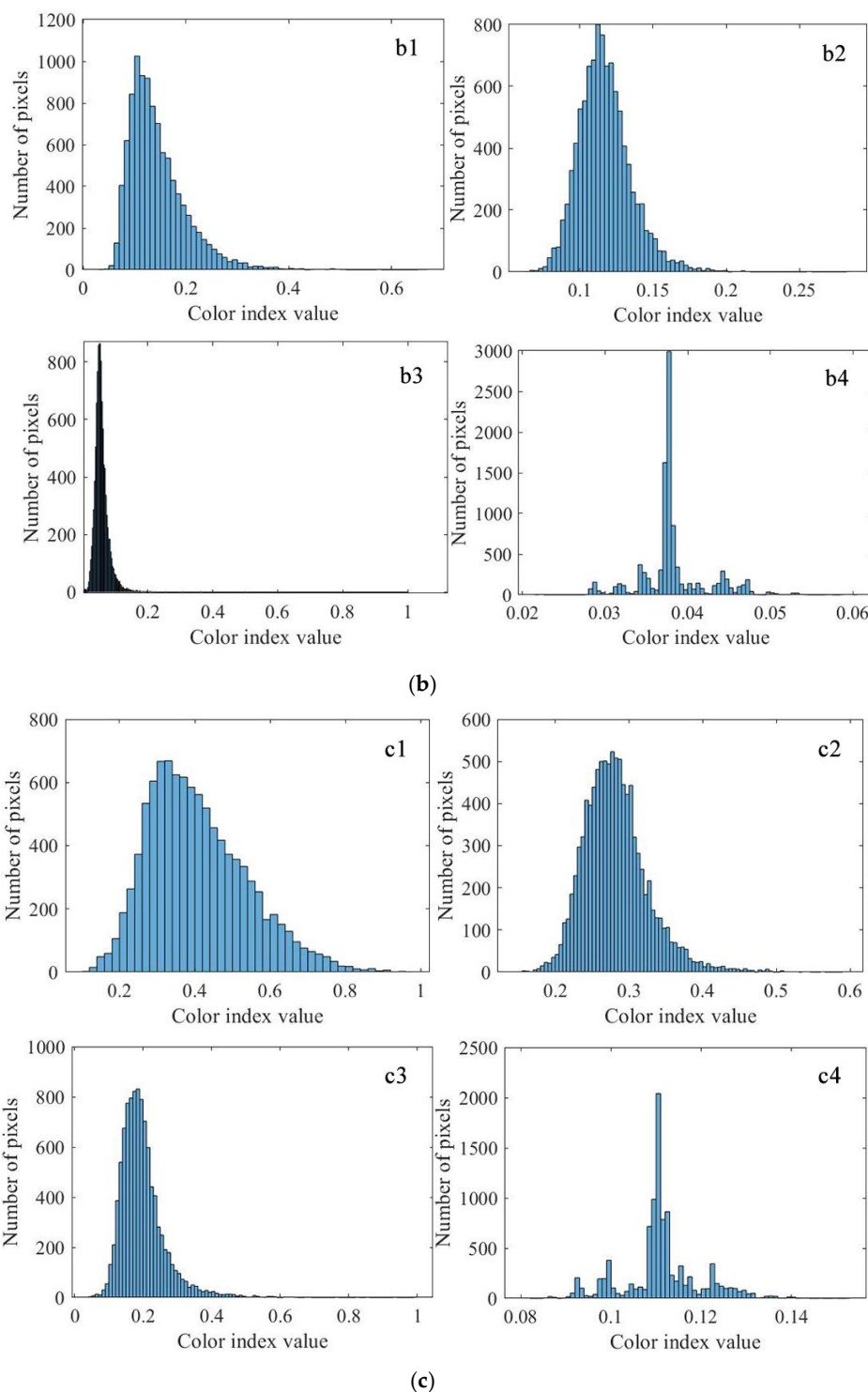

**Figure 5.** Histogram of the distribution of the 3 color indices. (**a**). NRBDI distribution histograms ((**a1**): Unharvested, (**a2**): Harvested, (**a3**): Tilled, (**a4**): Irrigated). (**b**). NGBDI distribution histograms ((**b1**): Unharvested, (**b2**): Harvested, (**b3**): Tilled, (**b4**): Irrigated). (**c**). MRBDI distribution histograms ((**c1**): Unharvested, (**c2**): Harvested, (**c3**): Tilled, (**c4**) Irrigated).

The analysis using Equation (5) shows that the algorithm can reduce the influence of a few extremes at both ends of the distribution. Equation (5) is used as the basis of this study for the classification of agricultural progress in farmland.

The characteristic values based on the color index and the model were analyzed by calculation of the distribution, the mean, and the standard deviation. In this study, the visual interpretation results were used as the actual values to verify the accuracy of the classification method.

The visual interpretation information was extracted by vectorizing unharvested wheat fields, harvested wheat fields, tilled fields, and irrigated fields in the UAV images based on ground survey data.

To verify the model developed in this paper, the classification accuracy was compared with the classification accuracies of other existing models. Three classification models, that is, random forest (RF) [41,42], support vector machines (SVM) [43], and maximum likelihood classification (MLC) [44], were used for this comparison.

These three classification methods were implemented using ENVI5.3. The classification results of the four methods were compared to verify the advantages of the method developed in this study.

*2.4. Accuracy Assessment*

In this study, we used the region of interest (ROI) (32.39° N and 119.42° E, in 2020) as a reference. This reference forms a comparison array with the number of pixels of each class in the classified image. The confusion matrix is then computed. The data of ROI were derived from the visual interpretation of high-resolution images of the study area. We considered the distribution area and complexity of different farming progress distributions in the rice–wheat rotation fields, and finally selected the following: eight unharvested ROIs with a total area of 2593.8 m$^2$; seven harvested ROIs with a total area of 2493.85 m$^2$; four tilled ROIs with a total area of 419.85 m$^2$; one irrigated ROI with a total area of 885.06 m$^2$. The validation samples of different farming progress types were obtained from the ROIs. We randomly selected the validation samples from the validation ROIs, and the number of samples for each farming progress type was 150. The samples of ground truth ROIs are the total ROI samples, including 4346 unharvested wheat samples, 2243 harvested fields samples, 228 tilled fields samples, and 1518 irrigated fields samples. The confusion matrix is a core component of accuracy evaluation [45]. It can not only describe the classification accuracy but also indicate the confusion among categories. The basic statistics for the error matrix were the product accuracy (Prod. Acc.), omission, and commission (Table 2). The Kappa coefficient indicates the degree of matching between the classification result and the real feature category [46]. The coefficient is used to evaluate the quality of classification. The formula is

$$K_{hat} = \frac{N \sum_{I=1}^{r} x_{ii} - \sum_{i=1}^{r}(x_{i+}x_{+i})}{N^2 - \sum_{i=1}^{r}(x_{i+}x_{+i})} \tag{5}$$

In the formula, *r* is the total number of columns in the error matrix; $x_{ii}$ is the number of pixels in row *i* and column *i* of the error matrix (i.e., the number of correctly classified pixels); $x_{i+}$ and $x_{+i}$ are the total numbers of pixels in row *i* and column *i*, respectively; *N* is the total number of pixels used for accuracy assessment.

**Table 2.** Error matrix statistics.

| Appellation | Formula |
|---|---|
| Prod. Acc | $p_{A_j} = p_{jj}/p_{+j}$ |
| Omission | $p_o = 1 - p_{ii}/p_{+j}$ |
| Commission | $p_c = 1 - p_{ii}/p_{i+}$ |

## 3. Results

*3.1. Color Indices Screening*

In the grayscale images, different agricultural progress types had different brightness responses, as shown in Figure 6. The selected indices could be screened by visual inspection. Normalization was applied to make the brightness reflection vary between 0 and 1. A higher

brightness yields a higher value [47]. The indices that could clearly distinguish different farming progress fields were NRBDI, NGBDI, and NGRDI. Therefore, these three indices were selected in this study for farming progress identification in rice–wheat rotation fields.

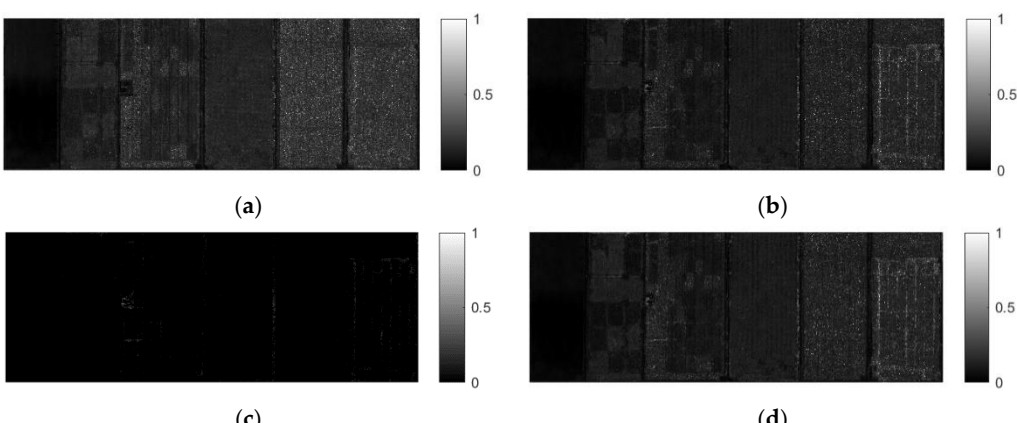

**Figure 6.** Grayscale images of four color indices. (**a**) NRBDI Chart. (**b**) NGBDI Chart. (**c**) NGRDI Chart. (**d**) MRBDI Chart.

The distribution of characteristic values of NRBDI, NGBDI, and MRBDI was obtained from the samples of different types of farm progress fields selected by visual interpretation, as shown in Table 3. The overlap in distribution intervals was apparent for all four farm progress types. This was because the constituents were not homogeneous in the same type of farmland. The reasons included, for example, that after harvesting wheat fields, the degree of stubble left varied; the same field stubble and soil alternate to form a field phenotype. Thus, there were too many scattered points of classification to allow whole-field identification (Figure 6), let alone meet the needs of whole-farm operations in the field. Although there was much overlap in the index distribution ranges of farmland types, the mean values of each farmland type were significantly different, as shown in Table 3. The mean values of the three indices for unharvested wheat fields, harvested wheat fields, tilled wheat fields, and irrigated fields decreased gradually.

**Table 3.** Statistical characteristic values of color indices among different types of fields.

|  | NRBDI | | | NGBDI | | | MRBDI | | |
|---|---|---|---|---|---|---|---|---|---|
|  | Range | Average | SD | Range | Average | SD | Range | Average | SD |
| Unharvested | [0.055, 0.95] | 0.222 | 0.093 | [0.15, 0.93] | 0.153 | 0.071 | [0.11, 0.99] | 0.413 | 0.13 |
| Harvested | [0.069, 0.42] | 0.156 | 0.033 | [0.058, 0.31] | 0.116 | 0.023 | [0.14, 0.70] | 0.303 | 0.059 |
| Tilled | [0.031, 0.59] | 0.109 | 0.037 | [−0.0038, 0.45] | 0.0676 | 0.025 | [0.062, 0.84] | 0.215 | 0.068 |
| Irrigated | [0.025, 0.094] | 0.0507 | 0.0076 | [0.0083, 0.0627] | 0.0305 | 0.0061 | [0.049, 0.18] | 0.1009 | 0.015 |

NRBDI: normalized red–blue difference index; NGBDI: normalized green–blue difference index; MRBDI: modified red–blue difference index.

The above analysis showed that the agricultural progress of the four types of rice–wheat rotation fields could not be effectively distinguished using only the index results. Therefore, models needed to be constructed to reduce the scattering of points in the classification results.

### 3.2. The Results of Model Construction

The formulas listed above were used to calculate the sample values, and the results are shown in Table 4. It can be seen from Table 4 that this method could effectively distinguish the color indices of unharvested wheat fields, harvested wheat fields, tilled fields, and irrigated fields. The distribution range of agricultural progress types in Table 4 was more discriminating than that in Table 3.

**Table 4.** Statistical eigenvalues based on color indices and model.

| Progress | NRBDI + RM | | | NGBDI + RM | | | MRBDI + RM | | |
|---|---|---|---|---|---|---|---|---|---|
| | Range | Average | SD | Range | Average | SD | Range | Average | SD |
| Unharvested | [0.194, 0.252] | 0.222 | 0.015 | [0.15, 0.93] | 0.153 | 0.071 | [0.11, 0.99] | 0.413 | 0.13 |
| Harvested | [0.144, 0.169] | 0.156 | 0.0079 | [0.058, 0.31] | 0.116 | 0.023 | [0.14, 0.70] | 0.303 | 0.059 |
| Tilled | [0.099, 0.122] | 0.109 | 0.0075 | [−0.0038, 0.45] | 0.0676 | 0.025 | [0.062, 0.84] | 0.215 | 0.068 |
| Irrigated | [0.0014, 0.0923] | 0.0507 | 0.0294 | [0.0083, 0.0627] | 0.0305 | 0.0061 | [0.049, 0.18] | 0.1009 | 0.015 |

NRBDI + RM: normalized red–blue difference index combined with regional mean model; NGBDI + RM: normalized green–blue difference index combined with regional mean model; MRBDI + RM: modified red–blue difference index combined with regional mean model.

According to the data shown in Table 4, we attempted to set the threshold values of the model (Table 5).

**Table 5.** Model threshold value range statistic.

| Progress | NRBDI + RM | NGBDI + RM | MRBDI + RM |
|---|---|---|---|
| | Range | Range | Range |
| Unharvested | [0.17, 0.252] | [0.135, 0.177] | [0.33, 0.451] |
| Harvested | [0.122, 0.17] | [0.08, 0.135] | [0.24, 0.33] |
| Tilled | [0.099, 0.122] | [0.06, 0.08] | [0.193, 0.24] |
| Irrigated | [0.0014, 0.099] | [0.0023, 0.06] | [0.0028, 0.193] |

NRBDI + RM: normalized red–blue difference index combined with regional mean model; NGBDI + RM: normalized green–blue difference index combined with regional mean model; MRBDI + RM: modified red–blue difference index combined with regional mean model.

### 3.3. The Accuracy of Model Construction

Table 6 shows the validation accuracy of three classification methods for the classification of different agricultural progress types. The validation samples were 150 for each type of agricultural progress. The highest accuracy of unharvested wheat fields was the MRBDI combined with the RM model: the Prod. Acc. was 96.7%, the omission was 3.3%, and the commission was 3.4%. The highest Prod. Acc. of harvested fields was the NGBDI combined with the RM model. The Prod. Acc. was 88.7%, the omission was 11.3%, and the commission was 28.8%. The highest Prod. Acc. of tilled fields was the MRBDI combined with the RM model: the accuracy was 79.3%, the omission was 20.7%, and the commission was 29.3%. The Kappa coefficients obtained with the NRBDI, NGBDI, and MRBDI combined with the RM model were 0.86, 0.82, and 0.88, respectively. The overall high Kappa coefficient proves that this research method is reasonable.

**Table 6.** Classification accuracy statistics of validation samples based on color indices and model.

| Progress | NRBDI + RM | | | NGBDI + RM | | | MRBDI + RM | | |
|---|---|---|---|---|---|---|---|---|---|
| | Prod. Acc. | Omission | Commission | Prod. Acc. | Omission | Commission | Prod. Acc. | Omission | Commission |
| Unharvested | 94.7% | 5.3% | 10.0% | 90.0% | 10.0% | 8.0% | 96.7% | 3.3% | 3.4% |
| Harvested | 87.3% | 12.7% | 20.7% | 88.7% | 11.3% | 28.0% | 88.0% | 12.0% | 7.5% |
| Tilled | 77.3% | 22.7% | 2.7% | 68.0% | 32.0% | 2.0% | 79.3% | 20.7% | 29.3% |
| Irrigated | 100% | 0% | 7.3% | 100% | 0% | 15.3% | 100% | 0% | 5.3% |
| Kappa | 0.86 | | | 0.82 | | | 0.88 | | |

NRBDI + RM: normalized red–blue difference index combined with regional mean model; NGBDI + RM: normalized green–blue difference index combined with regional mean model; MRBDI + RM: modified red–blue difference index combined with regional mean model.

### 3.4. Accuracy Assessment of Model Prediction Results

The types of agricultural progress in the study area were classified by constructing indices and models (Figure 7). Visual interpretation results were obtained by field investigation. The Prod. Acc., omission, and commission classification results of the three methods

are shown in Table 7. The highest accuracy of unharvested wheat fields was the NRBDI combined with RM model: the Prod. Acc. was 97.3%, the omission was 2.7%, and the commission was 4.3%. The Prod. Acc. of MRBDI combined with RM model was 96.9%, the omission was 3.1%, and the commission was 3.4%. The highest Prod. Acc. of harvested fields was the NGBDI combined with the RM model. The Prod. Acc. was 89.8%, the omission was 10.2%, and the commission was 19.9%. The highest Prod. Acc. of tilled fields was the MRBDI combined with the RM model: the accuracy was 78.2%, the omission was 21.8%, and the commission was 29.3%. MRBDI combined with the model could effectively reduce the dark patches in the blue area compared with NRBDI, i.e., it could effectively reduce the area of tilled farmland that was misclassified as irrigated fields (Figure 7).

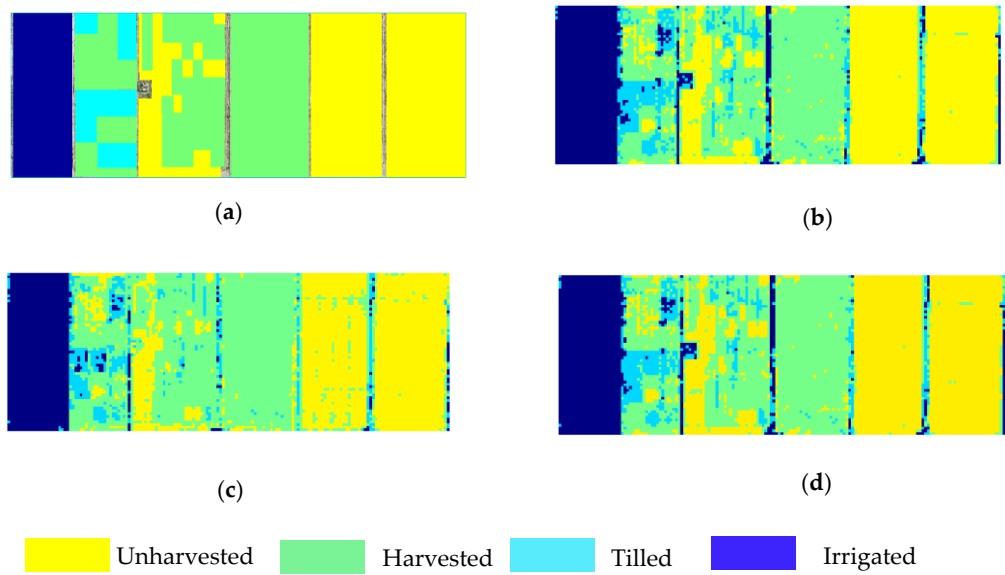

**Figure 7.** Classification result testing. (**a**) Visual interpretation result. (**b**) NRBDI classification result. (**c**) NGBDI classification result. (**d**) MRBDI classification result.

**Table 7.** Statistics of classification accuracy based on color indices and model.

| Progress | NRBDI + RM | | | NGBDI + RM | | | MRBDI + RM | | |
|---|---|---|---|---|---|---|---|---|---|
| | Prod. Acc. | Omission | Commission | Prod. Acc. | Omission | Commission | Prod. Acc. | Omission | Commission |
| Unharvested | 97.3% | 2.7% | 4.3% | 91.7% | 8.3% | 3.8% | 96.9% | 3.1% | 3.4% |
| Harvested | 87.6% | 12.4% | 6.7% | 89.8% | 10.2% | 19.9% | 89.0% | 11.0% | 7.5% |
| Tilled | 76.8% | 23.2% | 29.8% | 69.9% | 30.1% | 16.5% | 78.2% | 21.8% | 29.3% |
| Irrigated | 100% | 0% | 0.03% | 100% | 0% | 3.7% | 100% | 0% | 3.2% |
| Kappa | 0.9134 | | | 0.8738 | | | 0.9179 | | |

NRBDI + RM: normalized red–blue difference index combined with regional mean model; NGBDI + RM: normalized green–blue difference index combined with regional mean model; MRBDI + RM: modified red–blue difference index combined with regional mean model.

### 3.5. Performance Comparison with Other Models

In this study, a combination of color indices and the RM model was used to classify and identify the major farmland progress types in rice–wheat rotation fields. We compared the classification results with those from the random forest, support vector machine, and maximum likelihood methods (Figure 8). The samples of ground truth ROIs were selected by visual interpretation and comprised 4346 samples from unharvested wheat fields, 2243 samples from harvested fields, 228 samples from tilled fields, and 1518 samples from irrigated fields. The confusion matrix was obtained by comparing the ground truth ROIs with the classification results, and the Prod. Acc., omission, commission, and Kappa coefficients of four farming progress types were calculated based on the confusion matrix (Table 8). The Kappa coefficient in this paper reached above 0.9, indicating that the method constructed in this study was more stable for the classification of agricultural progress in

rice–wheat rotation fields. From the classification effect, the method of this study had the highest accuracy, where the extraction accuracy for unharvested wheat fields was 96.9%, for harvested wheat fields was 89.0%, for tilled fields was 78.2%, and for the irrigated fields was 100%. The Prod. Acc. of the random forest model was the lowest, with respective values of 62.8%, 87.3%, 71.4%, and 95.1%. The method described in this paper was more effective than the traditional machine learning classification methods in extracting the farming progress types that were consistent with actual farming operations.

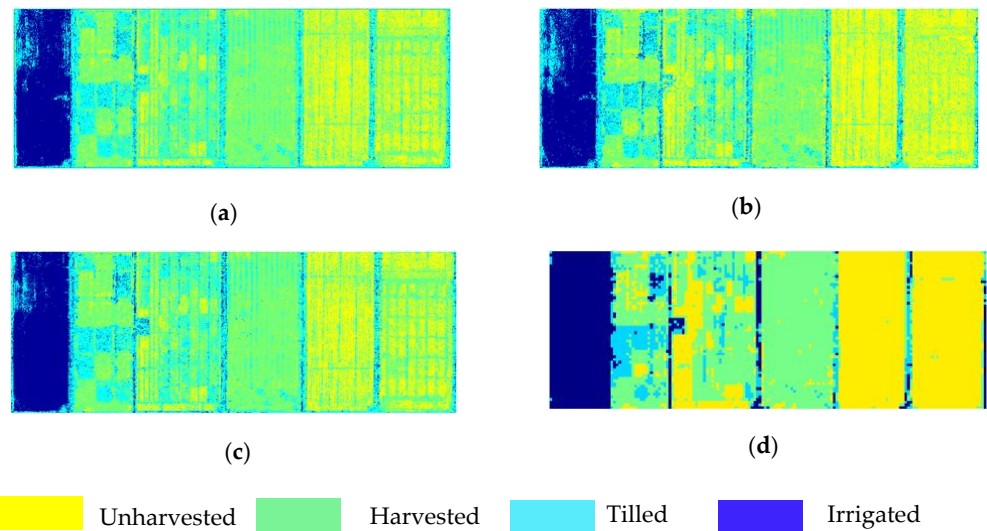

**Figure 8.** Comparison of classification results. (**a**) SVM. (**b**) MLC. (**c**) RF. (**d**) Method of this article.

**Table 8.** Statistics of extraction accuracy of each classification method.

| Methods | Unharvested | Harvested | Tilled | Irrigated | Kappa |
|---|---|---|---|---|---|
| SVM | 62.5% | 89.7% | 72.0% | 97.9% | 0.6467 |
| MLC | 77.5% | 80.1% | 71.0% | 95.9% | 0.6688 |
| RF | 62.8% | 87.3% | 71.4% | 95.1% | 0.6337 |
| MRBDI + RM | 96.9% | 89.0% | 78.2% | 100% | 0.9170 |

SVM: support vector machines; MLC: maximum likelihood classification; RF: random forest; MRBDI + RM: modified red–blue difference index combined with regional mean model.

## 4. Conclusions and Discussion

A classification method combining color indices and the regional mean model was proposed for analyzing the types of agricultural progress types in rice–wheat rotation farmland. The extraction and classification of the farm progress information of rice–wheat rotation in the study area were successfully realized. We verified the model accuracy, and the Kappa coefficients obtained by combining the NRBDI, NGBDI, and MRBDI with the RM model were 0.86, 0.82, and 0.88, respectively. Furthermore, by comparing the results of cropland information from different classification methods, determining the optimal combination of features, and comparing the results with traditional machine learning classification methods, we demonstrated that the method of color indices combined with the RM model was the best for farmland classification, with 96.9% extraction accuracy for unharvested wheat fields, 89.0% extraction accuracy for harvested wheat fields, 78.2% extraction accuracy for post-harvest tilled fields, and 100% extraction accuracy for irrigated fields. The method employed in this study was more accurate than the traditional machine learning classification methods in identifying the different farming progress types in the field, which should help to target the whole-field operation for different types of farming progress.

We proposed a new approach for the UAV-based classification of agricultural progress in rice–wheat rotation fields. Overall, the method was successful in achieving accurate

classification of unharvested, harvested, tilled, and irrigated fields. We found that the MRBDI classification method combined with the RM model achieves the classification of the whole-field area of the rice–wheat rotation fields, thereby eliminating the "pretzel" effect occurring in traditional machine learning classification, which was favorable to the machine for the next farming operation of the whole-field. The method also overcame the difficulty of distinguishing between unharvested fields mixed with rice-stubble and post-harvested rice-stubble fields. We believe that this classification method can be a new technique for monitoring the agricultural progress of rice–wheat rotation fields.

In addition, the method provided new insights and enhancement to farmland and crop classification. Its distinctive feature was that it can be applied to a wide range of rice–wheat rotation fields, overcoming the difficulty of distinguishing between the soil in rice-stubble interstices and tilled fields, thus achieving an accurate classification without removing the interference of soil background. This method also provided ideas for the extraction of rice–wheat rotation fields and UAV image preprocessing. Since the method can be applied to the classification of rice–wheat rotation fields, we believe it can avoid artificial field surveys and save human and material resources. Furthermore, the method avoided shelving farmland, thereby improving the efficiency of farm production.

While the results obtained in this study were highly accurate, there is still room for improvement. First, this study presented a classification method combining color indices and the RM model. The results obtained may be further improved if parameters such as texture and luminance are incorporated. Second, when collecting farmland information, the UAV images were collected under clear skies and low wind speed conditions between 11:00 am and 13:00 pm of local time. It remains to be further verified whether this method applies to other weather conditions, such as cloudy days. Third, this study was aimed at the classification of farming progress of farmland in the rice-stubble wheat region, concentrating on the wheat harvest and rice planting stages. Other stages were not classified, and thus the application of this method in monitoring the whole growing stages of wheat and rice was not verified. Fourth, this study area, therefore, the classification accuracy for a large area of farmland, remains to be verified.

**Author Contributions:** C.S., T.L., and C.J. conceived and designed the experiments; X.S. and C.S. performed the experiments; X.S., F.W., and C.S. analyzed the data and wrote the original manuscript; F.W., X.L., T.Y., C.J., and T.L. reviewed and revised the manuscript. All authors have read and agreed to the published version of the manuscript.

**Funding:** This research was funded by the National Natural Science Foundation of China (32172110, 32001465, 31872852), the National Key Research and Development Program of China (2018YFD0300805), the Priority Academic Program Development of Jiangsu Higher Education Institutions (PAPD), the Yangzhou University International Academic Exchange Fund Program, and the Jiangsu Creation Program for Post-graduation Students, China (KYCX21_3240).

**Institutional Review Board Statement:** Not applicable.

**Informed Consent Statement:** Not applicable.

**Data Availability Statement:** Not applicable.

**Acknowledgments:** We thank LetPub (www.letpub.com, accessed on 2 December 2021) for its linguistic assistance during the preparation of this manuscript.

**Conflicts of Interest:** The authors declare no conflict of interest.

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
