# Peer review of "The Classification of Farming Progress in Rice–Wheat Rotation Fields Based on UAV RGB Images and the Regional Mean Model"

_agriculture, doi:10.3390/agriculture12020124_

Round 1

Reviewer 1 Report

Dear authors,

I’ve read your paper focusing on the extraction of the farm progress information of rice-wheat rotation using UAV data. The use of several spectral indices, based on RGB data, and the comparison with other canonical classification model show how the model, and the adopted approach, is effective for the purpose of this paper. Additionally, I have appreciated that authors combine very simple optic data and specific classification modules usually related to more complex (multispectral or hyperspectral) remote sensed data. Nowadays, UAVs are very important instrument and show an important sprawl on their use on precision agriculture using, generally, more complex instrumentation. This paper show that, also using simple RGB dataset, useless information can be extracted and can be efficient for farmers to improve specifical agricultural monitoring practices.

I suggest only few modifications which are listed below.

Remarks:

  • In section 2, line129-132: I suggest providing a better explanation of how the process for image conversion to reflectance data was adopt.
  • The numeration of tables should be adjusted, and figures captions must be formatted as show in the guide for authors
  • Line 234: More information about the dimensionality, their spatial localization and the number of validation ROIs used should be done.
  • Line 317: why the samples of each validation dataset are so different? When a validation process is done an equal number of samples should be used (homogeneous in number and spatial positioning). The weight of each class should be similar.

Author Response

  1. In section 2, line129-132: I suggest providing a better explanation of how the process for image conversion to reflectance data was adopt.

Response: We have modified it to better express our meaning according to your suggestion.

  We decided to use the digital number (DN) of the RGB channel. The DN values of RGB channels in remotely sensed RGB images can quantitatively reflect the radiance and reflectance characteristics in visible spectrum of crop canopy [38]. Normalized DNs (r, g and b, Equations (1) – (3)) were reported to have superior capacity for vegetation estimation compared to original RGB DN values [39]. (Detailed information could be seen in Page 5 Lines 185-189.). Therefore, we decided not to use reflectance data but to use Normalized DNs for color index calculation. We are very sorry for the wrong description. We provide a better explanation of how the process for image in detail. We have revised the article. The content is given as follow: Following the processing procedure of Lu et al. [37]; image alignment, georeferencing, mosaicking, dense point cloud construction, orthoimage generation, and calibration were conducted using PhotoScan Professional software (Agisoft LLC, ST. Petersburg, Russia) and ENVI/IDL software (Harris Geospatial Solutions, Inc., Broomfield, CO, USA) based on the GPS location and camera internal parameters from each aerial image.

The related information has been modified in section 2.1. (Detailed information could be seen in Page 4 Lines 151-156.).

  1. The numeration of tables should be adjusted, and figures captions must be formatted as show in the guide for authors.

Response: We are very sorry that there is something wrong with our format. We have modified it in the article. Thank you very much for your suggestions. This information has been added in the 2.2 section. (Detailed information could be seen in Page 5 Lines 181; Page 10 Lines 294; Page 5 Lines 307;).

  1. Line 234: More information about the dimensionality, their spatial localization and the number of validation ROIs used should be done.

Response: We are very sorry that the loss of this information. We supplemented the dimensionality, and their spatial localization and the number of ROIs. And we reinterpreted the number of validation samples and the number of prediction samples. Thank you very much for your advice. The content is given as follow: In this study, we used the region of interest (ROI) (32.39°N and 119.42°E, in 2020) as a reference. (Detailed information could be seen in Page 9 Lines 260). We considered the distribution area and complexity of different farming progress distribution in the rice-wheat rotation fields, and finally selected the following: eight unharvested ROIs with a total area of 2593.8 m2; seven harvested ROIs with a total area of 2493.85 m2; four tilled ROIs with a total area of 419.85 m2; and one irrigated ROI with a total area of 885.06 m2. The validation samples of different farming progress types were obtained from the ROIs. We randomly selected the validation samples from the validation ROIs, and the number of samples for each farming progress type was 150. The samples of ground truth ROIs are the total ROI samples, including 4346 unharvested wheat samples, 2243 harvested fields samples, 228 tilled fields samples, and 1518 irrigated fields samples. (Detailed information could be seen in Page 9 Lines 263-272.).

This information has been added into our manuscript. (Detailed information could be seen in Page 9 Lines 260, 263-272.).

  1. Line 317: why the samples of each validation dataset are so different? When a validation process is done an equal number of samples should be used (homogeneous in number and spatial positioning). The weight of each class should be similar.

Response: I am very sorry that we incorrectly wrote the number of samples of ground truth ROI into the number of verified samples. We strongly agree that the samples of each validation dataset should be equal, we have added the description of the number and accuracy of validation samples in section 3.2: Page 12 Line 330-345; and corrected the writing of the number of samples contained in the ground truth ROI. The reason why the samples of each type of agricultural progress contained in the ground truth ROI are different is that the distribution area of different agricultural progress types in the ROI is different. The aim of this paper is to find out a good classification method of crop progress in rice-wheat rotation farmland. Therefore, it mainly compares different classification methods. The samples of the same agricultural progress type among different classification methods are the same. Therefore, we believe that although the samples of different agricultural progress types are different, they will not affect the screening results of the optimal classification method in this paper. Thank you very much for your advice. (Detailed information could be seen in section 3.2: Page 12 Line 330-345; Page 14 Lines 370、373; Page 14 Line 391-393; Page 1 Line 20-22; Page 1 Line 25-27.).

We appreciate for Editors/Reviewers’ warm work earnestly, and hope that the correction will meet with approval.

Best regards!

Reviewer 2 Report

I read this manuscript with interest. Overall, the information is well presented, and the manuscript is easy to read. The manuscript presents technical and scientific merits and interesting results about classification of farming progress in rice-wheat rotation fields based on UAV RGB images. I believe that the Manuscript falls within the scope of this Journal.

The objectives are original. Title is adequate. Abstract and Introduction are good. Material and methods are suitable but can be improved. Results and Tables were well presented and have several important evaluations for the understanding of the proposed subject. Discussion is well focused and combined with the literature. Conclusion is pertinent and in accordance with the objectives of the Manuscript. References are current, related to the scope of the Manuscript. However, there is too much citation and Chinese authors, and in the world, there are several other authors that could have been used.

Below I present some questionsso that the authors can improve the article

Keywords: All keywords are listed in the title and therefore must be changed.

Line 47: What is the average size of properties in China. Detail better how small and fragmented these areas are.

Line 67: But if we're talking about small producers and small areas, is this low coverage really a problem?

Line 71: UAV remote sensing technology has only advantages? Why do the authors not comment on the disadvantages, such as flight autonomy?

Lines 99-110: the hypothesis and objectives of the work are not sufficiently established.

Line 118: The UAV is battery powered with autonomy for how many minutes?

Line 118: I think that the information in the area, , such as geodetic coordinates, should be better described and informed in the text. What is the soil type?

Line 119: Which cultivars? What is the vertical and horizontal precision?

Line 146 and Figure 1: What were the conditions that differentiated the images presented for each area?

Line 191: What was the cirterium for this selection?

Line 372: It's important to prove it. What were the climatic variables at the time of data collection?

References: 80% of the references are from Chinese authors. There are several other works in the world.

Author Response

  1. Keywords: All keywords are listed in the title and therefore must be changed.

Response: Thank you very much for your advice. We have changed some keywords so that they are not all listed in the title; The content is given as follow: Keywords: UAV RGB image; Regional Mean model; Color index; Rice-wheat rotation field; Farming progress; Classification; Precision agriculture; (Detailed information could be seen in Page 1 Lines 32-33.).

  1. Line 47: What is the average size of properties in China. Detail better how small and fragmented these areas are.

Response: We are very sorry for the lack of information description. We add the average size of properties in China. And we detail better how small and fragmented these areas are. The content is given as follow: Rice-wheat rotation field is one of the main farming methods in China, with 4.8 million hectares of land under cultivation [5]. However, Chinese farmland areas are relatively small and fragmented [6]. In the study of the relationship between farmland fragmentation and agricultural production cost, many scholars use the average plot area, the number of plots and the average plot distance to quantitatively reflect farmland fragmentation [7–9]. According to statistics, in China, the average arable land area of each household is only 0.58hm2, the number of plots in each household is as high as 5.34, the average plot area is only 0.11hm2, and the average distance from home is 1.06km [10]. (Detailed information could be seen in Page 2 Lines 49-57)

References

  1. Yang, Y.; Ding, Q.; Zhao, Y.; Sun, C.; Wang, F. Optimization of the rotary tillage tool for wheat strip-till planter. Journal of South China Agricultural University. 2021, 42, 110-115.
  2. Ning, X.; Yu, X. Crop Rotation, Agricultural Planting Structure and Sustainable Food Security in China. Inquiry into economic issues 2018, 78–88.
  3. Tan, S.; Heerink, N.; Kruseman, G.; Futian, Q.U. Do Fragmented Landholdings Have Higher Production Costs? Evi-dence from Rice Farmers in Northeastern Jiangxi Province, P.R. China. China Economic Review 2008, 19, 347–358.
  4. King, R.; Burton, S. Land Fragmentation: Notes on a Fundamental Rural Spatial Problem. Progress in Human Geogra-phy 1982, 6, 475–494.
  5. Bentley, J. Economic And Ecological Approaches To Land Fragmentation: In Defense Of A Much Maligned Phenome-non. Annual Review of Anthropology 1987, 16, 31–67.
  6. Zhang, B. Theory, model and route on spatial recombination of arable land use system based on land fragmentation perspective. PhD Thesis, China Agricultural University, 2017.

  1. Line 67: But if we're talking about small producers and small areas, is this low coverage really a problem?

Response: We agree with the reviewer that low coverage is not a big problem for this paper, so we make a change in line 74-76 The content is given as follow: Although the ground platforms can provide images with high resolution, their relatively low working efficiency cannot meet the requirements of fast and efficient monitoring of farming progress. (Detailed information could be seen in Page 2 Lines 74-76.).

  1. Line 71: UAV remote sensing technology has only advantages? Why do the authors not comment on the disadvantages, such as flight autonomy?

Response: We are very sorry for the lack of information description. We complement the disadvantages of UAV remote sensing. The content is given as follow: UAV remote sensing platforms include fixed-wing and multirotor configurations. Fixed-wing UAVs have the advantages of fast flight, high flight efficiency, long endurance time, large payload capacity, and a high flight altitude [22,23]. However, they have certain requirements for taking off and landing, they cannot hover, and they will cause blurred images because of their high-speed shooting [24]. Multirotor UAVs have a simple struc-ture, the ability to hover, and modest requirements for taking off and landing, but they possess a slow flight speed, short endurance time, a low flight altitude, and small payload volume [24]. (Detailed information could be seen in Page 2 Lines 79-86.).

References

  1. Maria, T.; Reynolds, M.P.; Chapman, S.C. A Direct Comparison of Remote Sensing Approaches for High-Throughput Phenotyping in Plant Breeding. Front. Plant Sci. 2016, 7, 1131-.
  2. Valavanis, K.P.; Vachtsevanos, G.J. UAV Swarms: Models and Effective Interfaces. Springer Neth. 2015, 10.1007/978-90-481-9707–1, 1987–2019.
  3. Zhang, C.; Kovacs, J.M. The Application of Small Unmanned Aerial Systems for Precision Agriculture: A Review. Precis. Agr. 2012, 13, 693-712.

  1. Lines 99-110: the hypothesis and objectives of the work are not sufficiently established.

Response:

We are very sorry for not sufficiently describing our hypothesis and objectives of the work. We added this information. The content is given as follow: Furthermore, UAV remote sensing images contain high-throughput information that could be used to characterize changes in farming progress, but most investigations only used the high-throughput information to monitor crop growth, and few studies were oriented toward farmland research to monitor farming progress during harvest. Roth et al. [35] successfully predicted yield and protein content through the dynamics of soybean vegetative growth based on hyperspectral images. They provide strong evidence for UAV remote sensing in crop physiological analysis. But their research focused on the crop itself, with little mention of monitoring soybean fields to guide farm operations. Hassan et al [36]. realized the rapid monitoring of NDVI in wheat growth cycle by using the multi-spectral UAV platform, which can realize the prediction of grain yield. Their research focuses on the flowering stage and filling stage of wheat growth stage, but the growth and field monitoring of wheat harvest stage are seldom mentioned. Field monitoring during wheat harvest stage is important for guiding farmland operation of wheat harvest stage. This indicates the importance of conducting researches on farming progress monitoring in rice-wheat rotation fields based on UAV remote sensing platforms during the wheat harvest stage.

Therefore, the objective of this study was to explore the optimal classification method for crop progress classification of rice-wheat rotation fields at harvest stage. This classification method can effectively improve rice-wheat rotation farmland monitoring practices for farmers and provide a new idea for the visible image classification based on UAV. In this paper, we present a method to effectively extract information on farming progress using UAV RGB images and the RM model. The remainder of the paper is organized as follows. Section 2 describes the proposed methods and the corresponding materials being used. Section 3 describes the performance of the proposed method as well as the comparison with traditional machine learning methods. Section 4 concludes this paper and pro-vides further discussion. (Detailed information could be seen in Page 3 Lines 105-130.).

References

  1. Roth, L.; Barendregt, C.; Bétrix, C.; Hund, A.; Walter, A. High-throughput field phenotyping of soybean: Spotting an ideotype, Remote Sensing of Environment, 2022, 269. doi: https://doi.org/10.1016/j.rse.2021.112797.
  2. Hassan, M.A.; Yang, M.; Rasheed, A.; Yang, G.; Reynolds, M.; Xia, X.; Xiao, Y.; He, Z. A Rapid Monitoring of NDVI across the Wheat Growth Cycle for Grain Yield Prediction Using a Multi-Spectral UAV Platform. Plant Science 2019, 282, 95–103.
  3. Line 118: The UAV is battery powered with autonomy for how many minutes?

Response: We are very sorry for the lack of information description. The content is given as follow: The weather on the measurement day was sunny, and the UAV flew for a short time from 11:00 to 13:00 to ensure sufficient and stable sunlight. The UAV is battery powered with autonomy for about two hours.   (Detailed information could be seen in Page 3 Lines 135、139-140.).

  1. Line 118: I think that the information in the area, such as geodetic coordinates, should be better described and informed in the text. What is the soil type?

Response: We are very sorry for the lack of information description. We added information in area and the soil type. The content is given as follow: The test field is located in Wenhui Road Campus of Yangzhou University (32.39°N and 119.42°E, in 2020), with a total area of 3.76 hectares. The type of soil in this area is loamy soil. (Detailed information could be seen in Page 3 Lines 142-144.).

  1. Line 119: Which cultivars? What is the vertical and horizontal precision?

Response: We are very sorry for the lack of information description. We added information about cultivars and UAV parameters. The content is given as follow: The main crops in the tested fields in May and June were wheat (cultivars: Yangmai-16 and Yangmai-23) and rice (cultivars: Suxiu-867 and Naneng-9108). (Detailed information could be seen in Page 3 Lines 144- 145). The main technical parameters of the platform are as follows: hover accuracy: vertical: ± 0.5m, horizontal: ±1.5m; The UAV can fly for 23 minutes with a pair of batteries. (Detailed information could be seen in Page 3 Lines 133-135).

  1. Line 146 and Figure 1: What were the conditions that differentiated the images presented for each area?

Response: We are very sorry for the lack of description of this information. The content is given as follow: Through visual interpretation and field investigation, samples were randomly selected from four agricultural progress types, as shown in Figure 3. (Detailed information could be seen in Page 5 Lines 166-167).

  1. Line 191: What was the cirterium for this selection?

Response: We are very sorry for the lack of description of this information. The content is given as follow: By means of visual observation combined with field investigation, samples were randomly selected from each agricultural progress type to cover most features of the same agricultural progress type. Ten samples of unharvested wheat, harvested wheat, tilled, and irrigated fields were selected by visual inspection. (Detailed information could be seen in Page 7 Lines 214-216).

  1. Line 372: It's important to prove it. What were the climatic variables at the time of data collection?

Response: We are very sorry for the lack of description of this information. Thank you very much for your advice. We have supplemented the climate of the experiment by controlling a single variable. For the study of climate conditions with multiple variables, we plan to carry out the next experimental study in the future. The content is given as follow: When collecting farmland information, the UAV images were collected under clear skies and low wind speed conditions between 11:00 am and 13:00 pm of local time. It remains to be further verified whether this method applies to other weather conditions, such as cloudy days. (Detailed information could be seen in Page 15 Lines 426-429).

  1. References: 80% of the references are from Chinese authors. There are several other works in the world.

Response: Thank you very much for your suggestions. We have improved the references and the related content. (Detailed information could be seen in Page 16 Lines 448、472、497、512、515、518、520、525、527、529).
